# Exploring Freshwater Regimes and Impact Factors in the Coastal Estuaries of the Vietnamese Mekong Delta

**Vinh Hoa Dang [1,2], Dung Duc Tran [3,4,*], Thuc Bich Thi Pham [1], Dao Nguyen Khoi [5], Phuong Ha Tran [1] and Ninh Trung Nguyen [1]**

1   HCMC Institute of Resources Geography, Vietnam Academy of Science and Technology, Ho Chi Minh City 700000, Vietnam; dhvinh@hcmig.vast.vn (V.H.D.); ptbthuc@hcmig.vast.vn (T.B.T.P.); thphuong@hcmig.vast.vn (P.H.T.); trungninh.dhtl@gmail.com (N.T.N.)
2   Graduate University of Science and Technology, Vietnam Academy of Science and Technology, Ho Chi Minh City 700000, Vietnam
3   Center of Water Management and Climate Change, Vietnam National University Ho Chi Minh City, Ho Chi Minh City 700000, Vietnam
4   NTT Hi-Tech Institute, Nguyen Tat Thanh University, Ho Chi Minh City 700000, Vietnam
5   Faculty of Environment, University of Science, Vietnam National University Ho Chi Minh City, Ho Chi Minh City 700000, Vietnam; dnkhoi86@gmail.com
*   Correspondence: dungtranducvn@yahoo.com

**Abstract:** Freshwater resources make an essential contribution to the livelihoods of millions of local people in the coastal estuaries of the Vietnamese Mekong Delta (VMD). However, coastal freshwaters currently face numerous threats, not least (i) changing tidal dynamics due to sea level rise and (ii) changes in river regimes due to dam construction upstream. This research explores the evolution of freshwater regimes in these coastal estuaries. Using process diagrams, freshwater distributions are mapped and analyzed. Application of statistical methods provides insight into freshwater flow cycles and variations in water regimes upstream at various measurement points within the estuaries. A previously calibrated and validated hydraulic model is used to simulate drought-year scenarios and spatial changes in freshwaters over time. Findings indicate decreasing river discharges in the flood season, but increasing discharges in the dry season, due to the impacts of hydropower dams. In addition, the driest months are shifting earlier. From this data, we derive rules of thumb regarding freshwater distributions in the coastal estuaries of the VMD. These relate to (i) the boundary beyond which freshwater is always found; (ii) the boundary where freshwater appears daily; (iii) the start of the freshwater season; (iii) the boundary where freshwater appears until February and until April; (iv) the end of the flood season; and (v) the number of days without freshwater per year. The trends discerned will help local freshwater users and decision makers formulate forward-looking, flexible strategies for freshwater exploitation, while also providing avenues for further research.

**Keywords:** Mekong river; estuary; hydraulic modeling; freshwater process; tidal changes

---

## 1. Introduction

Coastal estuaries are among the world's most productive environments for aquaculture, agriculture and industry, while also being very densely populated [1,2]. More than half of the world's mega-cities lie within 50 km of the coast, and population densities here are 2.6 times greater than inland. This raises critical issues of water security in coastal areas [3,4]. Indeed, a key concern of our times is the efficient dist8ribution and use of freshwater resources in coastal areas, to conserve and restore brackish ecosystem services while also providing for the socio-economic needs of residents and economic activity [5].

Freshwater inflows are fundamental to the functioning of estuarine processes [6]. However, the spatial and temporal distribution of freshwater is undergoing major change in many coastal estuaries, due to a combination of issues such as sea level rise and changes in hydrological flow regimes caused by development of reservoir systems upstream, alongside local exploitation of freshwater [4,7–9].

The salinity of estuary waters is a result of an interaction between freshwater fluxes from upstream and saltwater from the sea, with salinity levels varying both vertically and horizontally [10]. Several studies indicate that under the influence of sea level rise tidal dynamics will become an even greater determinant of the temporal and spatial distribution of freshwater in coastal estuaries [2,11–14]. Tide is the main driving force pushing seawater into rivers, causing fluctuations of estuary salinity. From the upstream side, river flows push against tidal currents, resisting salinity intrusion while creating a brackish zone in the middle area. In the dry season, seawater usually makes its way farther inland throughout the estuaries, due to the lower river flows typically measured in this period. When river flows increase, floodwaters flush saline water back downstream in a process that provides various benefits to aquatic habitats [15].

The balance maintained by this cycle is very relevant to the Vietnamese Mekong Delta (VMD), though freshwater flows in this delta's estuaries have also been strongly influenced by changes both in flow regimes upstream and in downstream tidal movements. Upstream, flow patterns have become abnormally varied due to the construction of numerous dams [9,14,16,17]. The main streams of the Mekong River are now blocked by six mega-reservoirs, and 40 smaller reservoirs regulate flows on tributaries [18,19]. Moreover, that number of upstream reservoirs is set to double by 2030, according to the Ministry of Natural Resources and Environment of Vietnam. This will affect some 100 billion $m^3$ of Mekong River waters, accounting for 18% of the total annual flow. From the seaward direction, mean sea levels are rising at a rate of 3.18 mm per year, with a total rise of 120 mm predicted by 2030 [20].

Dams developed upstream have significant effects on flow regimes downstream. Impacts have been documented, for instance, from two mega-reservoirs in southwest China: the Xiaowan (constructed in 2010) and Nuozhadu (constructed in 2014) with a total capacity of 39,800 million $m^3$ [21]. These dams may divert water away from the main streams into local waterways without consideration for the effects downstream [22]. Upstream dams and reservoirs have reduced the amount of freshwater flowing to the coastal estuaries of the VMD. Having less freshwater for agriculture and domestic uses has motivated coastal inhabitants to exploit alternative freshwater resources, for example, groundwater [23–25]. Increased groundwater exploitation, combined with the influences of sea level rise and upstream dams and reservoirs, is changing freshwater distribution both temporally and spatially across the VMD.

Against this backdrop, this study posed two principal research questions: (i) How is the freshwater distributed in the coastal area of the VMD? (ii) How strong is the influence of the hydrological dynamics of upstream flows on the freshwater distribution downstream? These questions have not been fully answered in previous studies, which principally considered from analyzing the highest salinity values to determine the highest saline intrusion boundaries in order to explore prevention or mitigation solutions [12,16,20,26]. Previous studies, therefore, have not explicitly identified the boundaries of freshwater availability in months of the dry season for potential exploitation in the VMD coastal areas as well as have not investigated the influences of flow dynamics in the upstream delta on freshwater appearance downstream.

The current study sought to gain a deeper understanding of trends in the distribution of freshwater in the VMD, as a first step towards finding solutions to sustain brackish ecosystems here. It also sought to clarify how upstream flows have changed and the effects of these changes on freshwater availability in the estuaries. Understanding these changes is essential not only to maintain sufficient freshwater in estuary ecosystems but also to provide appropriate freshwater-exploitation alternatives instead of groundwater over-extraction for local communities in areas affected by salinization. By analyzing upstream flows and flows in the surrounding rivers, this study provides the first-ever mapping of freshwater distribution in the research area. We analyzed monitoring data from a number of upstream

and 18 coastal stations and simulated the spatial distribution of freshwater using a hydrodynamic model in order to seek answers to the research questions.

## 2. Study Area

The Mekong River runs through six countries, extending from China to Myanmar, Laos, Thailand, Cambodia and Vietnam. With an area of 40,000 km$^2$, the VMD is located in the most downstream region of the Mekong River basin (Figure 1). Hydraulically speaking, Kratie, Cambodia, can be considered the basin's upper boundary. Another boundary point, also in Cambodia, is where the Tonle Sap River flows into the Tonle Sap Lake. River waters flow into the lake during the rising and high stage of the annual floods, but flows are reversed during the falling stage and during the following dry season. The Tonle Sap Lake retains about 10% of the total wet season flow volume at Kratie, reducing the maximum discharge at Kratie by 16% [27,28].

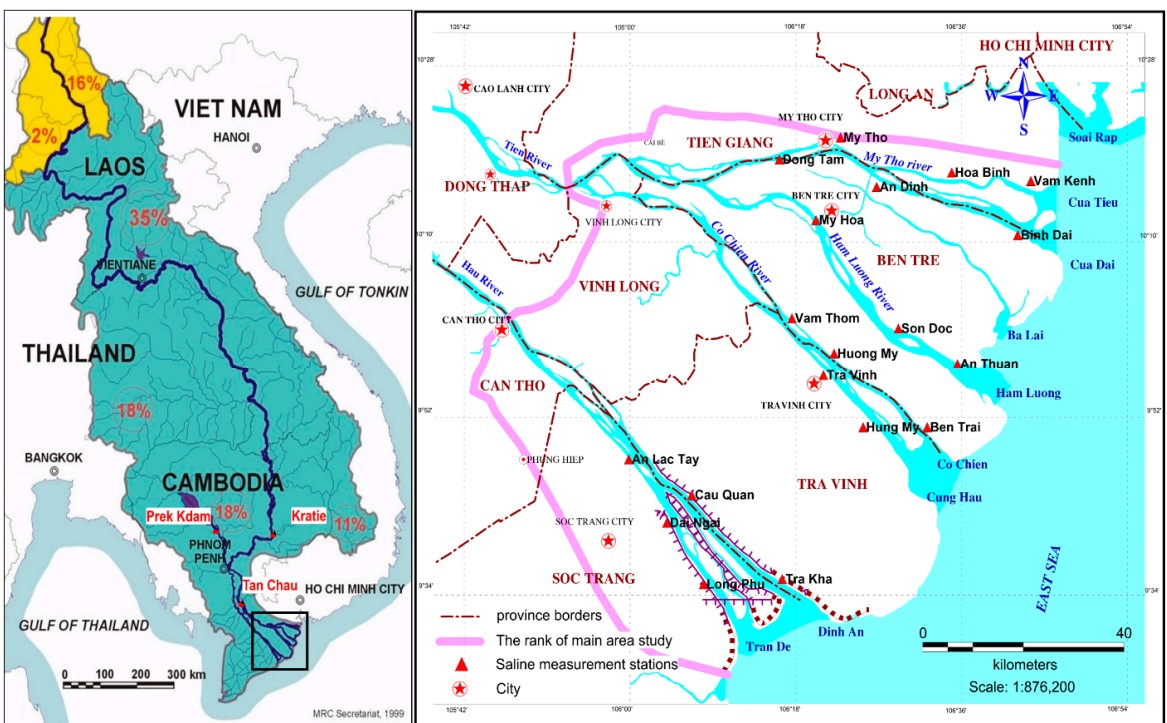

**Figure 1.** Location of the study area, with Kratie station and the Tonle Sap Lake at Prek Kdam, Tan Chau in the northeast of the delta (left) and the study's 18 monitoring stations (right).

From the Vietnam-Cambodia border, the Mekong River splits into two tributaries, called the Mekong (or Tien) River and the Bassac (or Hau) River, before flowing to the sea via nine river mouths in the Cuu Long region of Vietnam. This region supports a population of some 18 million and has a diverse economy including rice-based agriculture and aquaculture [5].

In combination with the river fluxes upstream, the coastal area of the VMD has been strongly influenced by tidal regimes. This has led to the formation of estuaries, defined as "a semi-enclosed coastal body of water, which has a free connection with the open sea, and within which sea water is measurably diluted with freshwater derived from land drainage" ([29], quoted in [30]: 11). As noted, the salinity of estuary water is a product of the interaction between freshwater from upstream and saltwater from the sea. Due to the complexity of such interactions, spatial and temporal distributions of estuarine freshwater vary, depending on the influences of the different surrounding flows.

The following are some key features of the salinity intrusion process in the study area:

At the river mouths of the VMD, there is a complex disturbance between the freshwater flows from the rivers upstream and the saltwater layer from the sea. The estuary is highly stratified. In the

rainy season, freshwater flows are quite high so the flow disturbance in the coastal area is usually weak. In this season, the flows are stratified and salinity regularly appears. In the dry season, freshwater flows from upstream are low, and the ratio of seawater to river water increases; therefore, the mixing is more complex.

On high tide days (the 1st–4th days or the 15th–19th days of every lunar month), the high tide peak is high and the lowest tidal water level is low; then seawaters entering the river mouth dominate. The mixing is strong and not stratified; hence, there is no saline wedge. On low tide days (the 7th–10th days or the 23rd–27th days of every lunar month), the high tide peak is low and the lowest tidal water level is high; then intrusion of seawaters into the river mouth decreases slightly. Depending on the strength of freshwater flows from upstream to the river mouth, a moderate or weak disturbance may occur. At these times, stratified flow can occur with the appearance of salty wedges.

## 3. Data Collection and Methods

Figure 2 presents the data and methodological framework used to explore the spatial distribution of freshwater in the coastal estuaries and the influence of upstream flows on freshwater regimes.

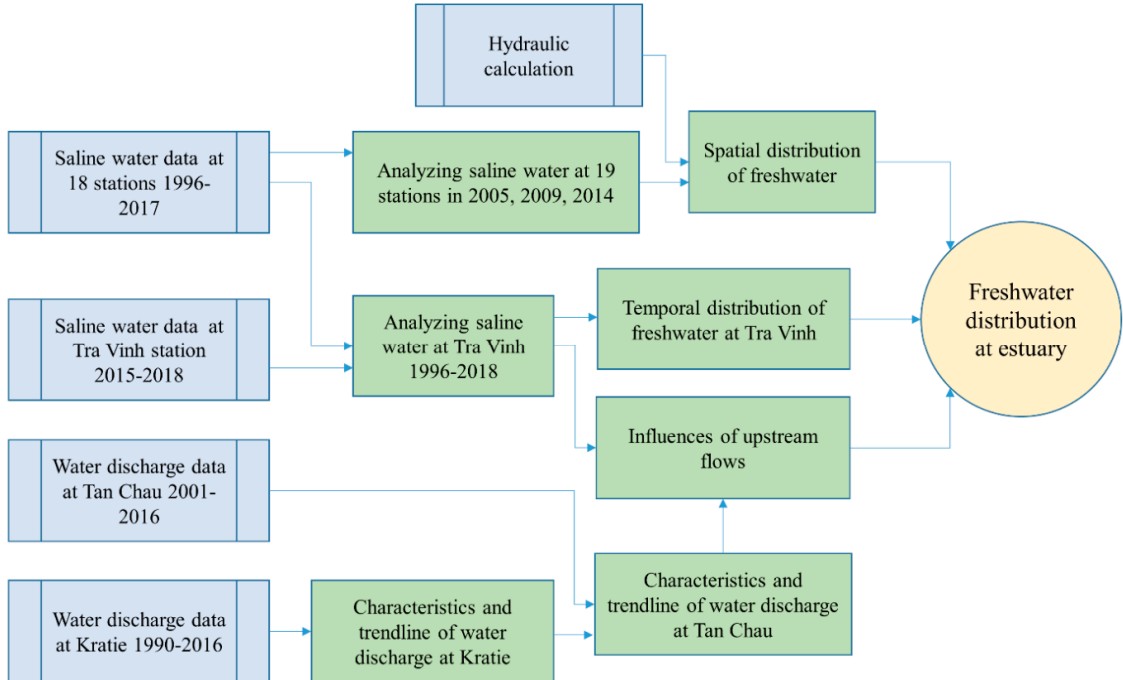

**Figure 2.** Methodological framework for simulating the spatial and temporal effects of upstream flows on freshwater distributions in the research area.

Our analyses used the following data: (i) Salinity measures from 1996 to 2017 from 18 monitoring stations throughout the VMD coastal region, obtained from the Hydro-Meteorological Data Center of Vietnam. These data were registered manually at a frequency of every 2 hour based on a specific schedule, equivalent to 12 data points at the odd hours of each day. Data were recorded during the dry season from February to June for the period before 2012. After 2012, data were recorded from January to June. (ii) Salinity and water level data were recorded automatically at 20-minute intervals over the 2015–2018 period at Tra Vinh station, located on the Co Chien River. These data were measured using WTW-LF 196 German conductivity meters with samples in glass bottles with a cork. At each measurement site, we collected the samples at three depths—0.2$H$, 0.5$H$ and 0.8$H$—with $H$ being the depth, which varied depending on the location (The average salinity (STT) is calculated using the following formula: $S_{TT} = (S_{0.2h} + 2 \times S_{0.5h} + S_{0.8h})/4$ (‰)). (iii) Water levels measured at Kratie station in the Mekong River for the period 1990–2016. This location is considered the starting point

of the Mekong Delta. It is 215 km from Phnom Penh and 310 km from the Vietnam border along the river. (iv) Flow monitoring data at the Tan Chau station of the Tien River for the 2001–2016 period. The Mekong River Commission provided these upstream data.

The data from the 18 monitoring stations provided a good indication of the spatial boundaries of salinity intrusion. The data series from 1996 to 2017 enabled us to identify the driest year as 2005. Further, 2009 was an average dry year, and 2014 was the least dry year. Analyses using the three-year data (2015–2018) then served to interpolate the data from the 18 stations, which helped us to identify the salinity boundaries. We considered three boundaries: the boundary where freshwater appears daily; the boundary where freshwater appears until February; the boundary where freshwater appears until April.

As the objective of this study was to obtain a better understanding of the spatial distribution of freshwater in the coastal area of the VMD, we could not use only the data measured at the 18 stations. To predict salinity allocations along the coast we employed a numerical 1-dimensional model, calibrated and verified using measurement data. To elaborate on the spatial distributions of the salinity boundaries and to provide further detail on freshwater zones, hydrodynamic modelling was done in MIKE 11. This is software developed by the Danish Hydraulic Institute [31]. We used the available MIKE 11 model, which the authors had calibrated and validated in previous studies [9], [26]. Due to the complex flow interactions in the VMD, we applied the unsteady mode for the model simulations. We set up the model using two modules: (i) the hydrodynamic (HD) module to simulate flow modes and (ii) the advection-dispersion (AD) module for salinity simulation. These modelling methods have been widely applied in studies of the Mekong Delta (see [9,12]).

Our computational network (or grid) included rivers and terrain from the lower Mekong region to the sea with the upper boundary at Kratie (see Figure 1). The grid comprised the following key components: (i) seven upstream flows with Kratie as the main boundary station; (ii) water levels at 68 boundary points, and salinity parameters determined from 10 coastal monitoring stations; and (iii) rain and evaporation data from 24 meteorological stations in the basin. In addition, Manning and dispersion coefficients were calibrated to adjust, respectively, upstream discharges and salt expansion parameters (Table A1 in the Appendix A presents some of the calibrated coefficients). Results of the hydraulic calculations were used to create freshwater distribution maps for the Mekong River estuaries.

Data from the Tra Vinh station were used to explore temporal distributions of freshwater in the coastal zone. For this, we applied statistical analyses to three variables: average annual discharge, average discharge during the flood season and average discharge during the dry season. The 2015–2018 data gave us a better idea of freshwater distribution rules in the flood season and availability of freshwater in the dry season. In addition, we analyzed the 1996–2017 Tra Vinh data to understand the characteristics of freshwater distribution throughout the coastal estuaries. Furthermore, we compared hydrological data from monitoring stations at Kratie and Tan Chau in the upper delta to data from Tra Vinh along the coast to assess the effects of water discharge from the upper delta on coastal freshwater distribution.

Table 1 presents the spatial and temporal criteria (based on observation, experience and literature review) applied in this study to identify freshwater distribution in the research area.

Although this study attempts to analyze freshwater boundaries over time as detailed as possible for exploitation, we are able to analyze data of February, April, and the dry season owing to the following reasons. First, the study only investigates the freshwater boundary from February due to the data availability. Observation data before 2012 were only measured from February, so this study could not analyze freshwater boundaries of the months before. Second, the freshwater boundary in April is considered because this is the most exhausting month in the period of monitoring data, also known as the most difficult month appearing freshwater. Finally, the boundary not having freshwater during the dry season is used to assess the most difficult level of freshwater in the estuary. Freshwater boundaries in March and May also have important to be analyzed. However, the freshwater boundary in March is in the middle of the February and April boundaries whereas the freshwater availability in May in the

estuary is higher than that in April but still lesser than in February. Hence the study does not consider measured data in March and May.

**Table 1.** Spatial and temporal criteria applied to identify freshwater distribution and boundaries.

| Criteria | Definition |
| --- | --- |
| Non-salinity boundary | At this boundary, the highest salinity level is always less than or equal to 0.3‰. While some research defines freshwater as water with less than 0.5‰ saline [29], we take the higher standard adhered to by the Vietnam Ministry of Health [32] |
| Daily freshwater boundary | Freshwater appears here at least once on any day |
| The boundary of freshwater in February, April and the dry season* | Freshwater appears here at least once, respectively, in February, April and the dry season |
| The non-freshwater boundary in the dry season | There is no freshwater here between February and April |
| The start date of freshwater | This is the first date on which freshwater makes its appearance at the end of the dry season with thereafter five consecutive days of freshwater observed and the situation remaining similar for the next 15 days. |
| The end date of freshwater | This date specifies the first of five consecutive days without any freshwater at the beginning of the dry season |
| The number of hours each month with freshwater | The total hours that freshwater appears within a month. |
| The largest number of days without freshwater | The greatest number of consecutive days of the year with no freshwater available |

\* This criterion was formulated based on the authors' long experience analyzing monitoring data from the coastal region of the Vietnamese Mekong Delta. Formulation of the criteria is clarified below.

## 4. Results and Discussion

### 4.1. Distribution of Freshwater in the Delta Estuaries

4.1.1. Temporal Distribution of Freshwater at Tra Vinh

Salinity data from Tra Vinh shows that freshwater usually did appear during the rainy season from 2015 to 2018. During the dry seasons of 2015, 2017 and 2018, the salinity level was less than 0.3‰ on many days. From 18 December 2015 to the end of January 2016, freshwater appeared only twice (29–30 December 2015 and 22–24 January 2016). In that season, the longest period without freshwater was 94 days (Figure 3).

The measurement data show variation in the appearance of freshwater at Tra Vinh from 1996 to 2017 (Figure 4). Freshwater began to appear from late May to early June, with the freshwater season usually lasting until February. There were seven years in which freshwater first appeared in May; 13 years in which freshwater first appeared in June; and one year in which freshwater first appeared in July. In saline months, freshwater was found an average of 116.1 h, 90.9 h, 40.3 h and 106.7 h, respectively, in February, March, April and May. The longest observed period without freshwater was in 2005, with 110 consecutive days, followed by 2016 with 94 days.

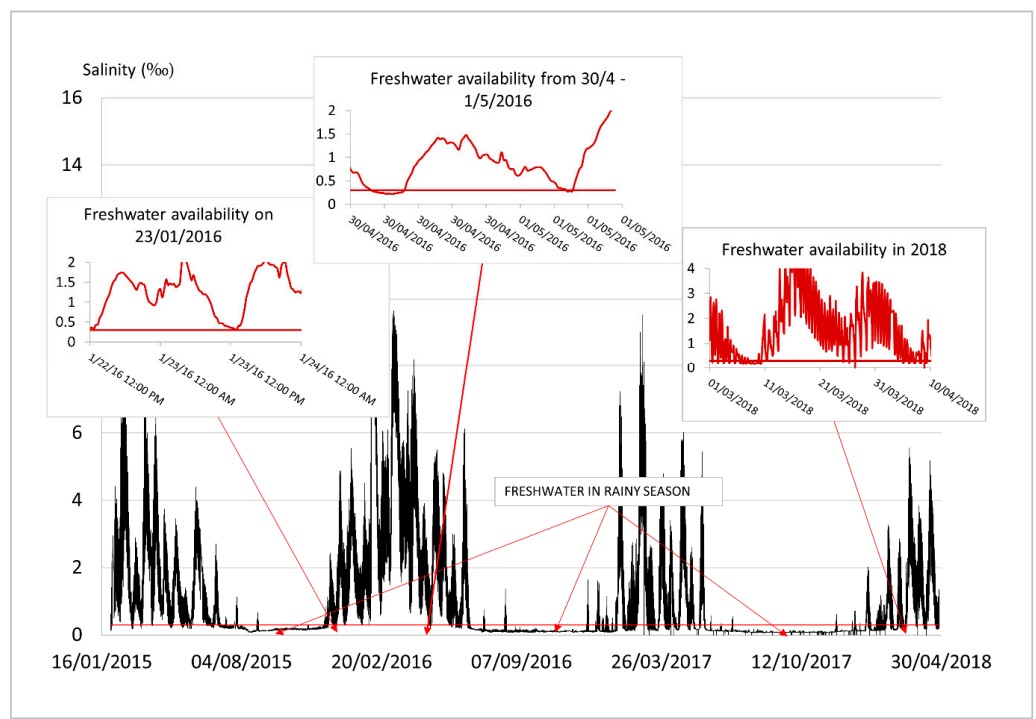

**Figure 3.** Salinity and appearance of freshwater at Tra Vinh, 2015–2018.

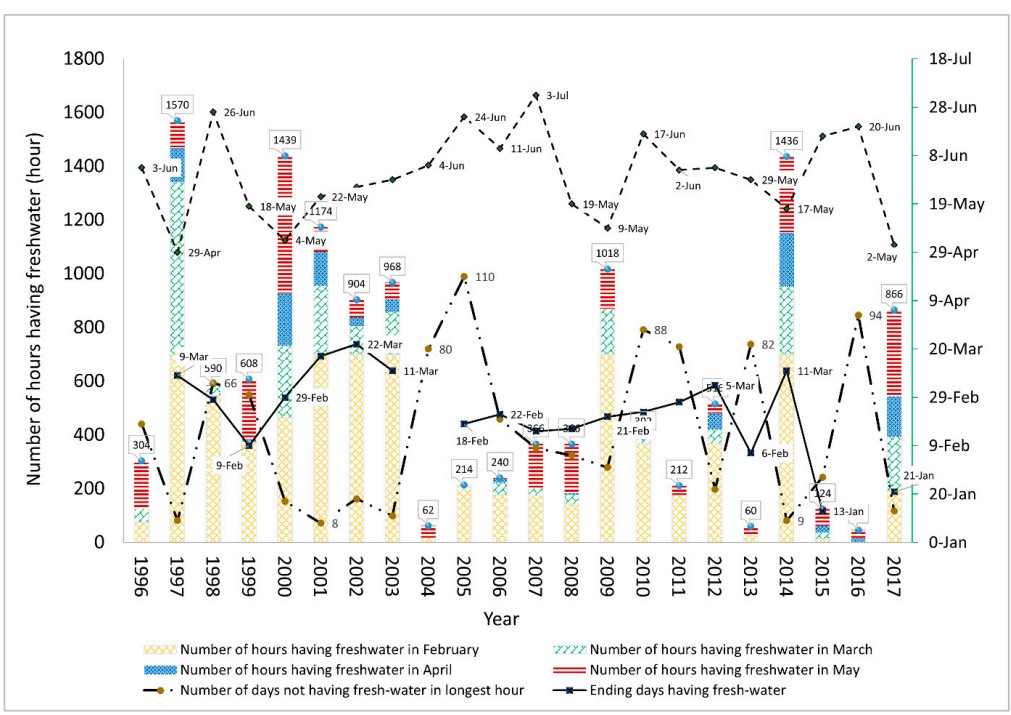

**Figure 4.** Statistics on freshwater at Tra Vinh, 1996–2017.

### 4.1.2. Spatial Distribution of Freshwater Based on Monitoring Data

The years 2005, 2009 and 2014 were selected to depict freshwater boundaries in the study area, as they were statically representative of years with the lowest, average and greatest amounts of freshwater, respectively. Figure 5 presents freshwater boundaries in February, April and daily, based on measurement station data.

*February freshwater boundaries*. Upstream flows tended to still be high in February, producing a rather stable freshwater boundary in the estuaries (Figure 5a). In 2005, the freshwater boundary in February was about 25 km from the sea. In 2009 and 2014, that distance was about 15 km.

*Freshwater boundaries in April*. The lowest upstream flows tended to be measured in April, which is why the freshwater boundary in this month is so close to the boundary of where freshwater is always available (Figure 5b). On the My Tho River, which is particularly sensitive to changes in upstream flows, the freshwater boundary in April was about 60 km from the sea. [12] and [7] similarly found salinity on the My Tho River to be sensitive to upstream flows.

*Daily freshwater boundaries*. In 2005, there was always freshwater on the Hau and Co Chien River, on the Ham Luong River and on the My Tho River, respectively, some 45 km, 55 km and 65 km from the sea (Figure 5c). In 2009, there was always freshwater on the Hau and Co Chien River at about 40 km from the sea. For the Ham Luong and My Tho rivers, these distances were, respectively, 45 km and 47 km from the sea. In 2014, the freshwater boundary was often observed 30 km from the sea during the dry season, and the trend was relatively stable. The boundaries fluctuated an average of about 20 km; that is, 15 km on the Hau and Co Chien River, 25 km on the Ham Luong River and 35 km on the My Tho River.

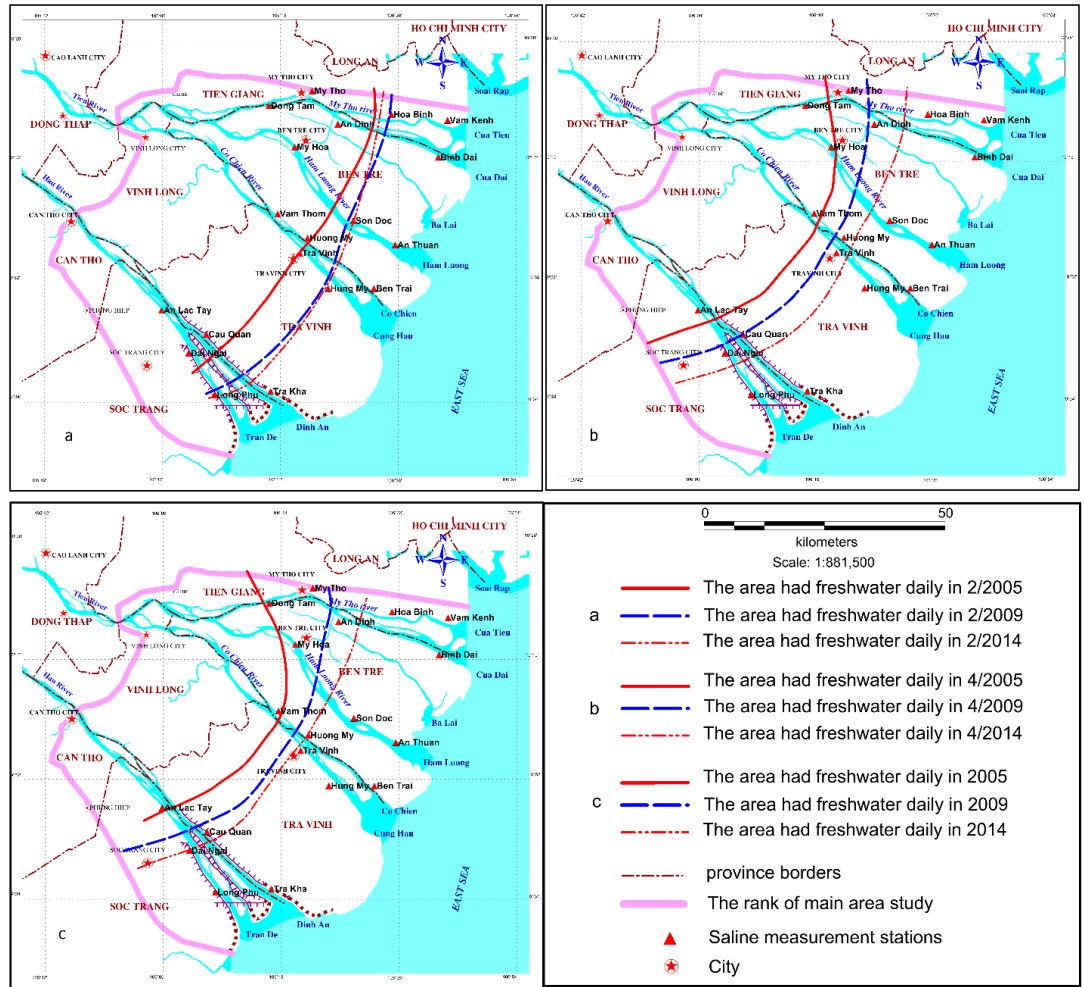

**Figure 5.** Freshwater boundaries estimated for 2005, 2009 and 2014. (**a**) The boundary where freshwater was always available in February moves further towards the sea; (**b**) the boundary where freshwater was always available in April moves further inland; (**c**) the boundary where freshwater was available every day moves further inland. Greater freshwater fluxes upstream (in 2014 and 2009 compared to the lowest water year of 2005) correspond with more freshwater appearing downstream, flushing saltwater to the sea.

### 4.1.3. Spatial Distribution of Freshwater Based on Hydrologic Modeling

Results from MIKE 11 models indicate five types of areas: (i) where there is always freshwater (non-saline areas), (ii) where there is freshwater daily, (iii) where there is freshwater until April, (iv) where there is freshwater until March and (v) where there is no freshwater in the dry season (Figure 6). The saline boundaries were found to be 80 km from the sea on the Tien River, 72 km on the Ham Luong River, 77 km on the Co Chien River and 62 km on the Hau River. The freshwater boundaries were 49 km (Tien River), 48 km (Ham Luong River), 38 km (Co Chien River) and 38 km (Hau River) from the sea. The simulation results were verified by our 2005 observation data.

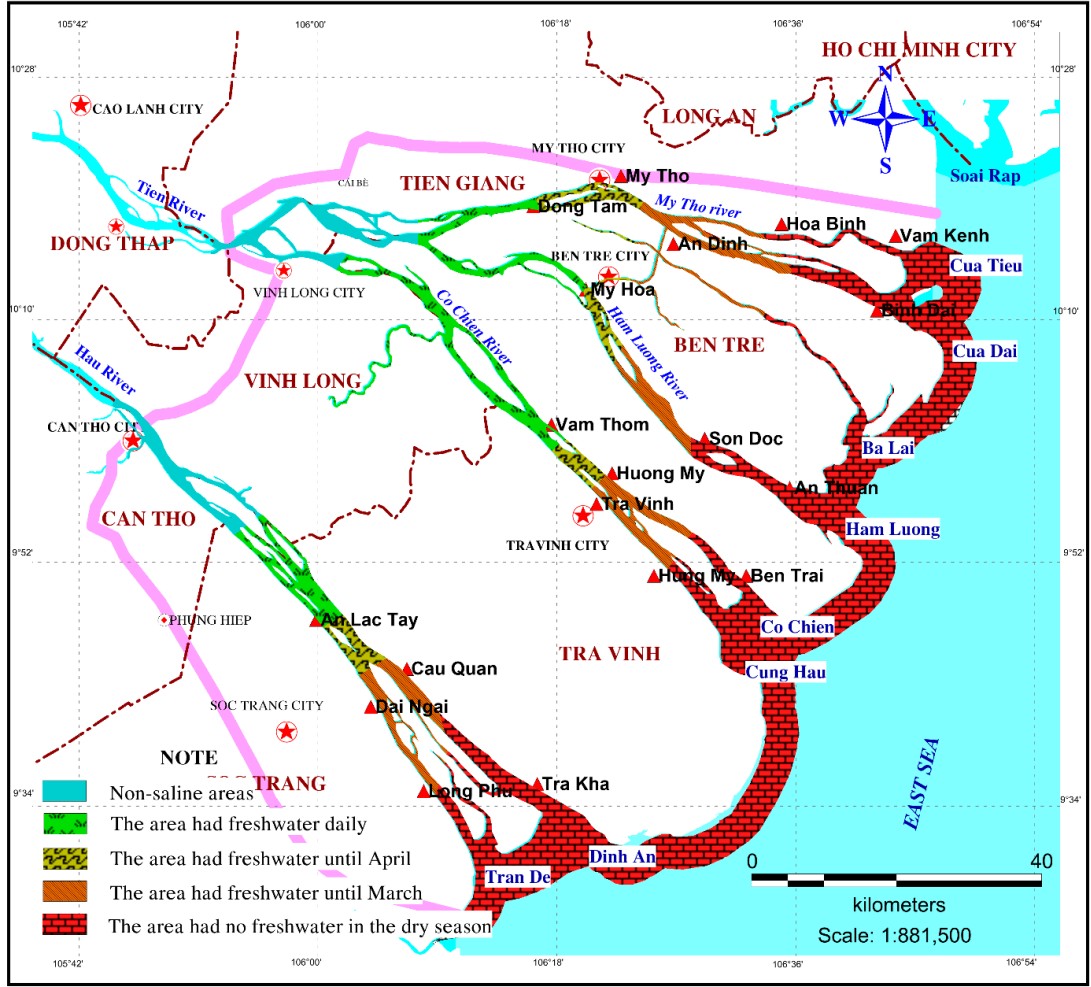

**Figure 6.** Freshwater zones in the dry season (January to May), 2005.

In general, the spatial distributions of freshwater from the modelling simulations were quite a good fit with the distributions indicated by the monitoring data (see Figure 6 compared with Figure 5, as well as the fitness presented in Table A1 of the Appendix A). Characteristics of the freshwater regime could thus be defined as follows: (i) The end date of the freshwater season usually occurred in February, though that date had recently shifted to January (2015, 2017), and once even occurred in December (2016); (ii) the starting date of the freshwater season was usually in late May or June. (iii) The longest period without freshwater at Tra Vinh was 110 days, in 2005, followed by 94 days in 2016. (iv) The hours with freshwater in the dry season (February, March, April and May) fluctuated sharply. The low was 46 h (2016); 60 h (2013) and 62 h (2004) were observed; the highest was 1570 h (2007). (v) Spatial boundaries were defined for non-saline (always freshwater) conditions, for freshwater appearing daily, for freshwater appearing until April, freshwater appearing until February and without freshwater at all in the dry season.



These characteristics impose the framework in which people living in the coastal zone must ration and plan their water use. The periods when inhabitants can access freshwater depend not only on these freshwater boundaries but also on the available storage capacity. The monthly hours of freshwater provide an indication of the amount of freshwater that can be added. In addition to the freshwater boundaries, the regularity and interval of freshwater availability thus determine the scale and feasibility of alternative methods of freshwater exploitation.

*4.2. Upstream Discharges and Changes Thereof*

4.2.1. Upstream Discharges at Kratie

Large fluctuations were observed in total annual discharges at Kratie, with the lowest discharge being only half the highest one. This is a disadvantage for downstream water users. Figure 7 shows the characteristics and trends of discharges at Kratie from 1990 to 2016. The annual average discharge was 12,214 m$^3$ s$^{-1}$ and the total annual discharge was about 385 billion m$^3$ year$^{-1}$. In the dry season, the total discharge was some 55 billion m$^3$, accounting for 14.3% of total annual flow. The highest and lowest annual discharges were 16,320 m$^3$ s$^{-1}$ and 7961 m$^3$ s$^{-1}$, respectively.

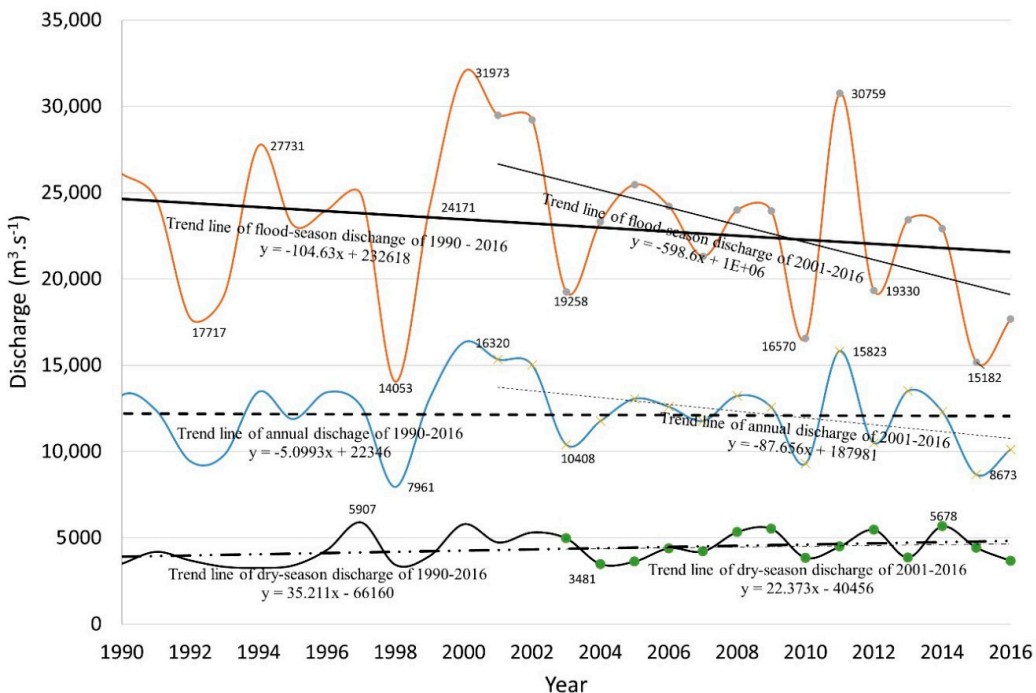

**Figure 7.** Trendlines of water discharges at Kratie, 1990–2016.

The trendlines in Figure 7 are divided into two periods, 1990–2016 and 2001–2016, based on the 2000 flood peak. The data in Figure 8 show an annual reduction of 243 m$^3$ s$^{-1}$ over the 2001–2016 period. In the flood season, the decrease was larger, with a reduction of 104 m$^3$ s$^{-1}$ for the whole observed period and 598 m$^3$ s$^{-1}$ for 2001–2016. Remarkably, the discharges in the dry season have tended to increase slightly, by about 35.21 m$^3$ s$^{-1}$ over the whole 1990–2016 period. These changes are primarily attributable to three factors, according to [7,9,33]: (i) the impacts of the upstream reservoir systems; (ii) increased exploitation of freshwater resources in the river basin; (iii) the effects of transferring water out of the catchments. The reservoirs reduce flows in the flood season and increase them in the dry season. The two other factors reduce flows throughout the whole year.

Examining the discharges in the dry season reveals the complex influences of the three factors (Figure 9). Although the discharges in the dry season increased slightly overall, the discharges decreased considerably in the first months of the dry season (in January and December). While [12]

and [7] concluded that the lowest discharges and the salt-intrusion peak were in April, our data show the lowest discharges in the dry season have shifted from April to March or even February. This means that saline intrusion in the Mekong Delta is coming sooner, posing a direct threat to winter-spring crops cultivated in the coastal provinces. Recent measurement data indeed show salinity peaks occurring in February (2013 and 2015–2017) or March (2014 and 2018).

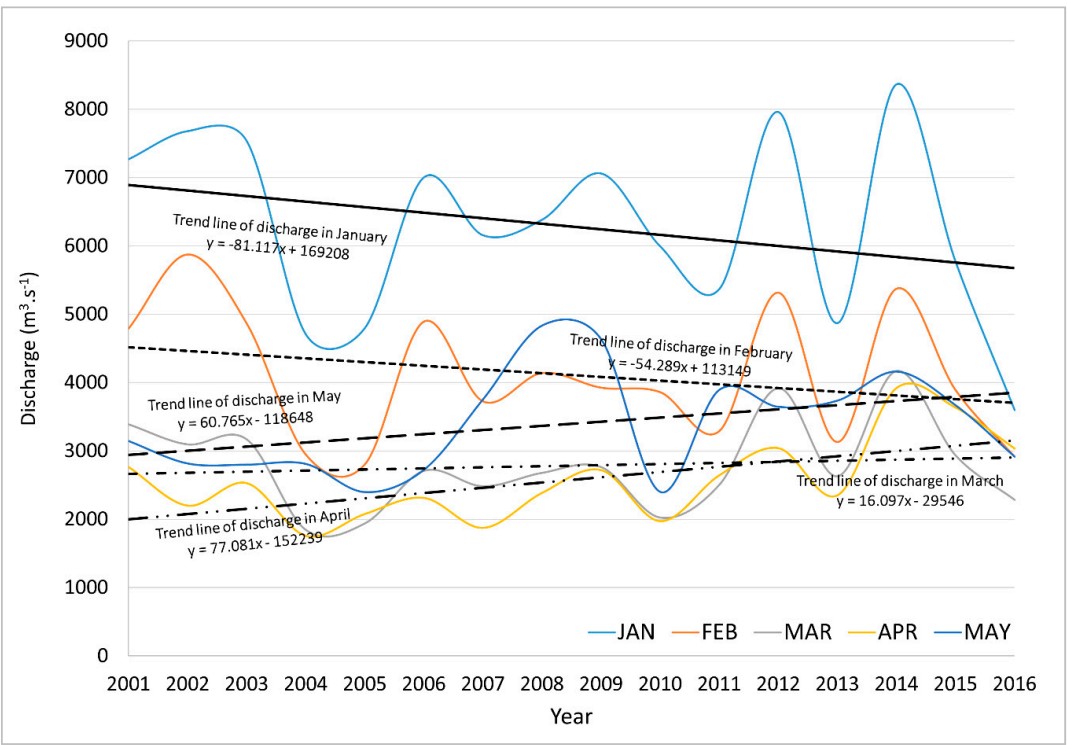

**Figure 8.** Trendlines of discharges at Kratie in the dry season (January to May), 2001–2016.

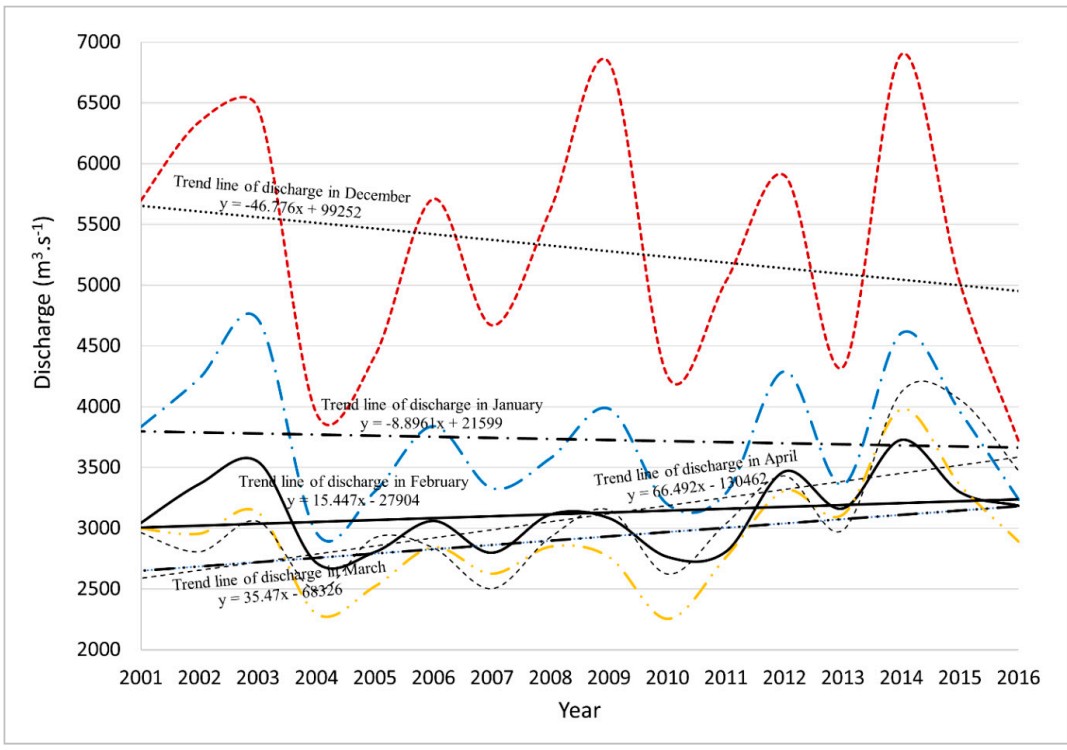

**Figure 9.** Trendlines of discharges at Tan Chau, 2001–2016.



4.2.2. Water Distributions from the Upper Delta to the VMD

Figure 10 compares flows at Kratie and Tan Chau. The correlation between both time series is quite tight. In deriving the correlation we accounted for the particularities of the flows from Cambodia. The Tonle Sap Lake (TSL) in particular plays a central role here, as also explored by [34,35]. In the dry season (December to April), the TSL contributes more than 20 billion $m^3$ of water, which is about half of the 40.3 billion $m^3$ that flows through Kratie. As flows from the TSL are concentrated in the early months of the dry season, we split flows from the TSL into two periods: (i) from January to the date on which flows from the TSL at the Prek Dam station exceeded 1000 $m^3$ $s^{-1}$ and (ii) the period after February, when the TSL discharge volume was less than 1000 $m^3$ $s^{-1}$. This produced a much closer relationship. The first phase of the relationship was established by the formula: $Q = 2.5703 \times Q_{Kratie} - 3439.7$ ($m^3$ $s^{-1}$). The second phase was calculated as $Q = 0.7035 \times Q_{Kratie} + 600.9$ ($m^3$ $s^{-1}$).

Toan etc [9] evaluated the correlation between the flows at Tan Chau and Kratie but did not consider the regulating effect of the Tonle Sap Lake, so the relationship they found was less tight.

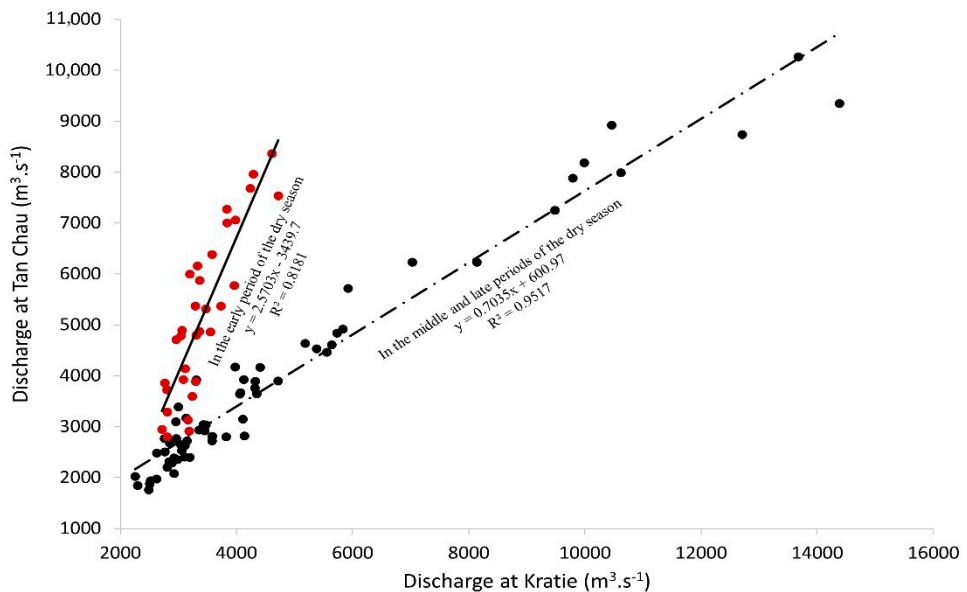

**Figure 10.** Correlation between monthly average flows at Kratie and Tan Chau in the dry season (January to May).

At the start of the dry season (in January and February), the discharges to Tan Chau are strongly affected by the TSL, which creates water fluxes at Tan Chau that are almost double those at Kratie (Figures 8 and 9). This relation is confirmed by the strong coefficient of determination ($R^2 = 0.82$) (Figure 10). However, later in the dry season (March, April and May), the contributions of the TSL are limited; therefore, the discharges to Tan Chau are driven directly by the discharge at Kratie. In those months, only 70% the discharge at Kratie, in the upper Mekong River, flow to Tan Chau; therefore, the correlation is quite high ($R^2 = 0.95$).

*4.3. Impact of Upstream Flows on Freshwater Distribution*

Figure 11a presents the relationship between the average flows in January at Tan Chau and flows at the end of the freshwater season in Tra Vinh. In general, the two flows are related, but not closely ($R^2 = 0.47$). When the average discharge at Tan Chau drops below 4273 $m^3$ $s^{-1}$, the freshwater season in Tra Vinh ends in early January, whereas when $Q$ in Tan Chau reaches 5424 $m^3$ $s^{-1}$, the freshwater season ends in early February.

Figure 11b presents the average flow for three months ($Q_{tb}$) in the dry season (April, May and June) at Tan Chau and at the beginning of the freshwater season. The relationship found here is closer

($R^2 = 0.72$). Accordingly, when $Q_{tb}$ is higher than 5500 m³ s⁻¹, the freshwater season starts in early May, and when the three-month $Q_{tb}$ is less than 3000 m³ s⁻¹, the freshwater season starts at the end of June. Figure 11 also presents the relationship between $Q_{tc}$ and the freshwater period at Tra Vinh. Overall the relationship is quite tight ($R^2 = 0.70$). This correlation suggests that when $Q_{tc}$ is greater than 4000 m³ s⁻¹, freshwater is available every month at Tra Vinh, and there are at least 88 hours of freshwater per month. According to the trendline, when $Q_{tc}$ reaches 7000 m³ s⁻¹, there is always freshwater at Tra Vinh. When $Q_{tc}$ is less than 2000 m³ s⁻¹, there is no freshwater at Tra Vinh. According to the trendline, when $Q_{tc} < 2761$ m³ s⁻¹, there is no freshwater at Tra Vinh. A month with less than 100 h of freshwater occurs only when $Q_{tc} < 4000$ m³ s⁻¹.

A relationship was found between average annual flows at Tan Chau from January to May and the longest period (in number of days) without the freshwater in the year (Figure 11c). Our results show that this relationship is quite tight ($R^2 = 0.80$). Accordingly, when the average flow at Tan Chau is less than 3200 m³ s⁻¹, the number of successive days on which there is no freshwater at Tra Vinh can be up to 80 days. When the five-month average flow at Tan Chau exceeds 5000 m³ s⁻¹, there is freshwater every day at Tra Vinh.

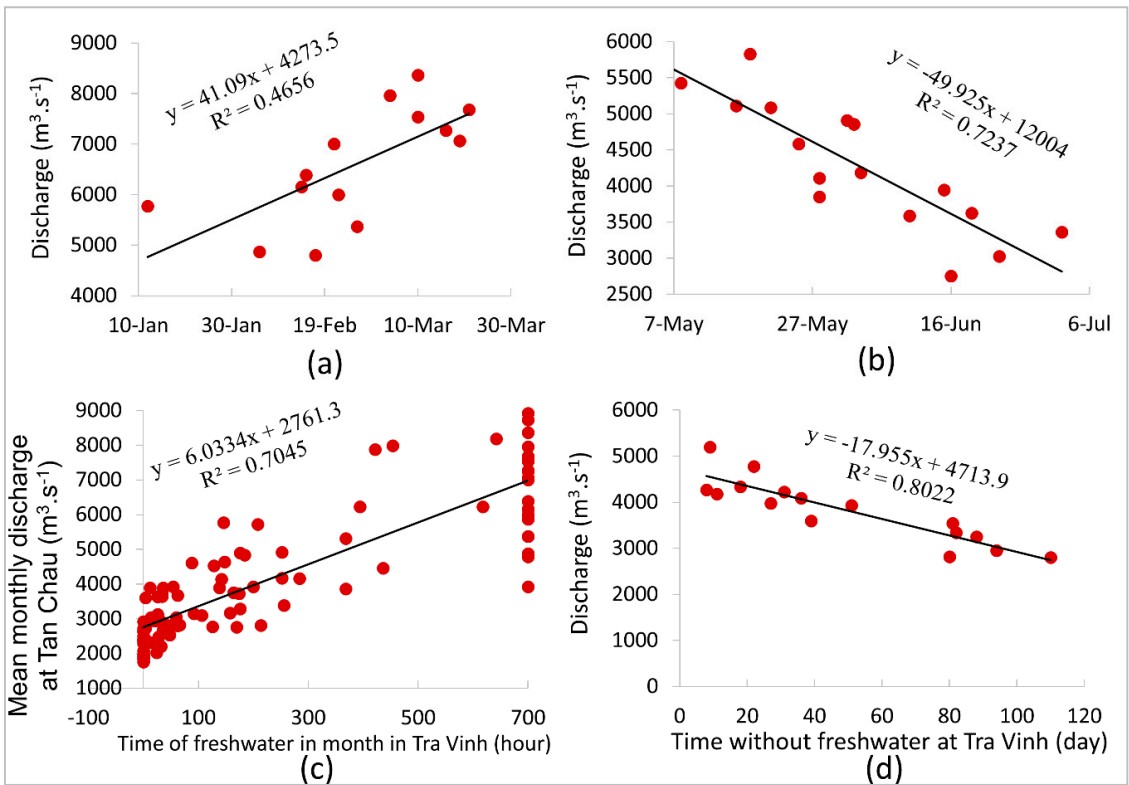

**Figure 11.** (**a**) Relationship between flows at Tan Chau from January to April and the timing of the late freshwater season at Tra Vinh. (**b**) The relationship between flows at Tan Chau in the three-month dry season (from April to June) and the timing of the start of the freshwater season at Tra Vinh. (**c**) The relationship between $Q_{tc}$ and the length of the freshwater period at Tra Vinh, the 700 value is used in the plots to indicate freshwater always being present in the dry season. (**d**) The relationship between mean annual flows at Tan Chau from January to May and the highest number of days without freshwater at Tra Vinh.

Water salinity in the estuaries is a result of an interaction between freshwater from upstream and saltwater intrusion from the sea. Freshwater boundaries thus fluctuate in space and time, depending on upstream river flows and those from the sea [7,12]. As tide is the main force pushing seawater into the rivers, salinity fluctuates with tidal rhythms [7,12]. To examine the impact of upstream flows on freshwater distribution on the coast, we looked at the relationship between flows at Tan Chau and at

Tra Vinh. We identified the starting and ending date of the freshwater season, the number of hours that freshwater was available in the dry season and the number of days without freshwater in the year.

Based on the identified correlations with upstream flows, we were able to make some predictions regarding freshwater flows in the estuaries (Figure 11d). Our findings suggest the following trends: (i) average flows at Tan Chau in January will continue to decrease, causing the end of the freshwater season to come sooner; (ii) flows in the three months of March, April and May will continue to increase, leading to earlier start of the freshwater season; (iii) average flows at Tan Chau in January and February will decrease, with average flows in March, April and May increasing, resulting in similar development in freshwater availability in those months; and (iv) dry season flows will increase, reducing the greatest number of days without freshwater.

## 5. Conclusions

This study analyzed and mapped the spatial and temporal distribution of freshwater in the estuaries of the Mekong River. Rules were derived for freshwater distribution, based on which appropriate freshwater exploitation regimes can be developed. Distribution details include the boundary beyond which freshwater is always present; the boundary at which freshwater is present on a daily basis; the boundary at which freshwater is present until April; the boundary at which freshwater is present until February; the beginning of the freshwater season; the end of the freshwater season; the number of days without freshwater per year; and the hours of freshwater each month.

We analyzed specifically the influences of flows at Kratie and the Tonle Sap Lake on water distribution in the VMD estuaries. Our findings in this regard reinforce and advance results from previous work. Our analyses indicate that under the impacts of hydropower dams and reservoirs, discharges in the flood season will decrease, but they will increase in the dry season. Our findings also point out that the driest period has shifted from April to March and even February. Our findings furthermore clarify the relationship between monthly average flows at Kratie and Tan Chau. These were found to be tightly correlated, disputing findings from previous studies that analyzed the monthly average flow relationship between the two stations without accounting for the particularities of flows from the Tonle Sap Lake.

This study also found a relationship between upstream flows (i.e., at Kratie and Tan Chau) and the freshwater regime in the coastal estuaries. Rules of thumb derived from our data could enable changes in freshwater distribution to be better predicted in the future. As such, the end of the freshwater season will likely come sooner rather than later; and the freshwater season will likely start at an earlier date. Furthermore, there will be fewer successive days without freshwater, and the number of hours with freshwater in the dry months (March, April and May) will increase.

Freshwater regimes in estuarine zones are relatively moderated, without sudden changes. Distributions depend mainly on tidal movements and flow trends in the upper delta. This study focused only on factors impacted by upstream flows. Elements originating from the sea, especially sea level rise due to climate change, were not considered in detail. In addition, besides meteorological factors, the demand for water use in the estuarine zone was not evaluated, though this too directly affects the freshwater resources in estuaries.

Finally, yet importantly, this study highlights the concern raised by [36] that it is necessary to transform the growing amount of data collected into information for users. We also prove that recurring and typical hydrodynamic processes of any area are possibly recognized by conveniently processing some selected field data together with data availability.

**Author Contributions:** V.H.D. developed the methodology and concept, and data analysis, and structured and wrote the first draft; D.D.T. reviewed literature, edited the manuscript and hydrographs, and corresponding; T.B.T.P. collected, analyzed data and did the modelling; P.H.T. analyzed data and created maps; N.T.N. collected and analyzed data, and did the modelling; D.N.K. reviewed the state-of–the-art and structure of the manuscript.

**Acknowledgments:** The authors thank the Vietnam Academy of Science and Technology for funding the research at the VAST06.3/16-17.

**Conflicts of Interest:** The authors declare no conflict of interest.

## Appendix A

**Table A1.** Observed and simulated salinity concentration values at the stations of Tra Vinh, Song Doc, My Tho and Tan An in the coastal region of the Mekong Delta (adapted from [26]).

| Station | Max. Salinity Concentration (%) | | | Average Salinity Concentration (%) | | |
| --- | --- | --- | --- | --- | --- | --- |
| | Simulated | Observed | Difference | Simulated | Observed | Difference |
| **Tra Vinh** | | | | | | |
| 10–13 February 2005 | 5.8 | 6.2 | 0.4 | 3.5 | 3.7 | 0.2 |
| 18–21 February 2005 | 9.5 | 8.3 | −1.2 | 5.5 | 6.2 | 0.7 |
| 26 February–1 March 2005 | 7 | 6.8 | −0.2 | 3.9 | 5 | 1.1 |
| 11–14 March 2005 | 10 | 9 | −1 | 5.75 | 6.5 | 0.75 |
| 29 March–1 April 2005 | 9 | 10 | 1 | 8 | 7.75 | 0.25 |
| **Song Doc** | | | | | | |
| 10–13 February 2005 | 11 | 10 | −1 | 6 | 8 | 2 |
| 23–26 February 2005 | 11 | 10 | −1 | 7.5 | 8.5 | 1 |
| 11–14 March 2005 | 13 | 16 | 3 | 12 | 12 | 0 |
| 29 March –1 April 2005 | 14 | 16 | 2 | 13.5 | 13 | −0.5 |
| **My Tho** | | | | | | |
| 10–13 February 2005 | 1.2 | 0.7 | −0.5 | 0.35 | 0.6 | 0.25 |
| 26 February–1 March 2005 | 1.1 | 1.1 | 0 | 0.6 | 0.75 | 0.15 |
| 3–8 March 2005 | 2.6 | 2.2 | −0.4 | 1.2 | 1.6 | 0.4 |
| 17–21 March 2005 | 3.5 | 3.1 | −0.4 | 1.95 | 2.2 | 0.25 |
| **Tan An** | | | | | | |
| 10–13 February 2005 | 5 | 4.5 | −0.5 | 3 | 3.25 | 0.25 |
| 26 February–1 March 2005 | 6.5 | 6.5 | 0 | 4.8 | 5 | 0.2 |
| 11–14 March 2005 | 6.8 | 6.5 | −0.3 | 5.25 | 5 | 0.25 |
| 19–21 March 2005 | 9 | 8 | −1 | 6 | 6.5 | 0.5 |

The model network includes some 900 rivers and branches, 25,900 water level nodes, and 18,500 discharge nodes; the Manning coefficients calibrated range from 0.022 to 0.032.

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
