# Peer review of "Exploring Freshwater Regimes and Impact Factors in the Coastal Estuaries of the Vietnamese Mekong Delta"

_water, doi:10.3390/w11040782_

Round 1

Reviewer 1 Report

The authors mainly showed freshwater changes in the lower Mekong River based on analyzing data and modeling results. However, the authors need to make clear: (i) what the major scientific questions they try to investigate are; (ii) why these questions are important; (iii) how their analyses contribute to answer the questions. The authors need to dig and discuss more on the connections between their data and modeling results. The authors need to specify the logic and significance for the criteria like “the boundary of freshwater in February, April…”.

Author Response

Dear reviewers,

The authors would like to thank for your valuable comments. Please find our responses in the documents attached.

On behalf of the author group

Dung Duc Tran

PhD in Water Management.

Reviewer 2 Report

to authors:

1) measures: line 122: you state you use salinity measures: what kind of measurements are they? point measures? at what depth? depth averaged measures? how are they obtained? this information is crucial in order to interpret their outcome.

2) numerical model:

a) why do you need a numerical model in this study? how is it used? is it used to predict salinity distribution along the river network, compliant with the measured data? or is it used to predict distribution when no data are available. In other words, is it a predictive model or an analytic model?

a) since the numerical model plays an essential role in the study authors should give more details on it: is it a 1d-network model? is it steady state? What is the spatial discretization of the rivers? are the roughness and diffusion coefficients uniform? Can you provide a table of the calibrated coefficients? H

b) how well does the model matches the measurements? Would a two layer model like the one described in [1] give better/worse results? Please comment.

3) Authors should give details on the features of the saline intrusion in the region. Is it a stratified estuary? is it well mixed? what is the maximum speed of the "tip" of the intrusion during high tide/low discharge? is it reasonable to be approximated by an arrested wedge (steady state)?

4) are the measured data publicly available? if so, please provide a link.

5) Figure 6: certain boundaries of the river are highlighted (mouth of Hau River). What does that stand for?

6) figure 11.(c): Why is there an accumulation of point at 700 hours?

[1] Prestininzi, P., A.Montessori, M.La Rocca, G.Sciortino "Simulation of Arrested Salt Wedges with a Multi-layer Shallow Water Lattice Boltzmann Model." Advances in Water Resources

Volume 96, 2016, Pages 282-289

Author Response

(The authors gave the same response as above.)

Round 2

Reviewer 1 Report

The authors need to make it clear why the questions have not been fully answered in previous studies in Introduction part. The authors need to specify what text (Line number) they have added in section 4 to clarify the relationship between the data and modeling results. The authors need to answer why they choose the freshwater boundaries in February, April, not other months and why these boundaries are important to answer their questions.

Author Response

Points: The authors need to make it clear why the questions have not been fully answered in previous studies in Introduction part. The authors need to specify what text (Line number) they have added in section 4 to clarify the relationship between the data and modeling results.

The authors need to answer why they choose the freshwater boundaries in February, April, not other months and why these boundaries are important to answer their questions.

Response points: The authors appreciated the valuable comments of the reviewer. We address your comments as follows:

Point 1: The authors need to make it clear why the questions have not been fully answered in previous studies in Introduction part.

Response point 1:

We agree with the reviewer regarding the comment. Text is added into the Introduction (lines 85-91 in the revised manuscript) to make clear why the research questions have not been fully answered in previous studies.

Point 2: The authors need to specify what text (Line number) they have added in section 4 to clarify the relationship between the data and modeling results.

Response point 2:

In section 4 (4.1.3), the authors highlight the relationship between data and modelling results. Please find the text from lines 267-277.

Point 3: The authors need to answer why they choose the freshwater boundaries in February, April, not other months and why these boundaries are important to answer their questions.

Response point 3:

In the revised manuscript, the authors clarify why the freshwater boundaries in February, April were investigated in this research, but not other months.

Please find our text modified by using explanation caption below Table 1 (from line 208).

“Although this study attempts to analyse freshwater boundaries over time as detailed as possible for exploitation, we are able to analyse data of February, April, and the dry season owing to the following reasons. First, the study only investigates the freshwater boundary from February due to the data availability. Observation data before 2012 were only measured from February, so this study could not analyse freshwater boundaries of the months before. Second, the freshwater boundary in April is considered because this is the most exhausting month in the period of monitoring data, also known as the most difficult month appearing freshwater. Finally, the boundary not having freshwater during the dry season is used to assess the most difficult level of freshwater in the estuary. Freshwater boundaries in March and May also have important to be analysed. However, the freshwater boundary in March is in the middle of the February and April boundaries whereas the freshwater availability in May in the estuary is higher than that in April, but still lesser than in February. Hence the study does not consider measured data in March and May”.

Round 3

Reviewer 1 Report

The authors can consider incorporating their response to Point 3 into Section 3 Data collection and methods.

Author Response

Open Review

English language and style

( ) Extensive editing of English language and style required 
( ) Moderate English changes required 
(x) English language and style are fine/minor spell check required 
( ) I don't feel qualified to judge about the English language and style 

Comments and Suggestions for Authors

The authors can consider incorporating their response to Point 3 into Section 3 Data collection and methods

Point 1: The authors can consider incorporating their response to Point 3 into Section 3 Data collection and methods

Response: The authors appreciated the valuable comments of the reviewer. We incorporate the paragraph below into Section 3 Data collection and methods in the revised manuscript (page 6, lines 207-218):

“Although this study attempts to analyse freshwater boundaries over time as detailed as possible for exploitation, we are able to analyse data of February, April, and the dry season owing to the following reasons. First, the study only investigates the freshwater boundary from February due to the data availability. Observation data before 2012 were only measured from February, so this study could not analyse freshwater boundaries of the months before. Second, the freshwater boundary in April is considered because this is the most exhausting month in the period of monitoring data, also known as the most difficult month appearing freshwater. Finally, the boundary not having freshwater during the dry season is used to assess the most difficult level of freshwater in the estuary. Freshwater boundaries in March and May also have important to be analysed. However, the freshwater boundary in March is in the middle of the February and April boundaries whereas the freshwater availability in May in the estuary is higher than that in April, but still lesser than in February. Hence the study does not consider measured data in March and May”.